

# Levels of toothache-related interests of Google and YouTube users from developed and developing countries over time

Matheus Lotto[1,*], Patricia E.A. Aguirre[1], Anna P. Strieder[1], Agnes F.P. Cruvinel[2] and Thiago Cruvinel[1,*]

[1] Department of Pediatric Dentistry, Orthodontics and Public Health, Bauru School of Dentistry, University of São Paulo, Bauru, São Paulo, Brazil

[2] Discipline of Public Health, School of Medicine, Federal University of Fronteira Sul, Chapecó, Santa Catarina, Brazil

* These authors contributed equally to this work.

Corresponding author
Thiago Cruvinel,
thiagocruvinel@fob.usp.br

## ABSTRACT

**Background.** The preferences of Web users can be influenced by distinct factors of populations. Hence, hypothetically, source-based patterns of health-related Web searches might differ between individuals from developed and developing countries, due to their distinct educational, social, economic, political, cultural, and psychosocial backgrounds. In this context, this study aimed to determine the trends of toothache-related searches performed on Google Search and YouTube, regarding differences between developed and developing countries.

**Methods.** This retrospective longitudinal study analyzed computational metadata on toothache-related interests of Internet users. Google Trends was accessed to obtain the monthly variation of relative search volume (RSV) of the topic "Toothache-Disease" on Google (G) and YouTube (YT) through 2008–2017. Autocorrelation and partial autocorrelation plots, ARIMA models, Kruskal–Wallis, Dunn's and $T$ tests were performed for evaluating trends, 12-month forecasts and the differences of annual ratios of YT/G searches between developed and developing countries, respectively ($P < 0.05$).

**Results.** Uptrends of RSVs were observed in both country groups over time, although 12-month forecasts tended to plateau. The volumes of searches were higher in developed countries in comparison to developing ones; however, this difference was not observed regarding Google searches performed between 2016 and 2017. Independently of country groups, the ratios YT/G remained relatively constant throughout the period, indicating a greater interest in toothache-related information available on Google.

**Conclusion.** In conclusion, toothache-related searches from Google and YouTube increased during the last decade. The preferences of Web users seemed to be influenced by the differences between developed and developing countries, such as the availability and penetration of the Internet, and education levels.

## INTRODUCTION

The spreading of information and communication technologies (ICTs) empowered individuals to obtain health knowledge and decide about their own conditions, including searches for alternatives to their demands. It is noteworthy that most Internet users are interested in health-related topics retrieved by digital platforms (*Aguirre et al., 2018*), such as search engine tools, video-sharing websites, and social media (*Hansen et al., 2018*; *Atkinson, Saperstein & Pleis, 2009*).

In this context, we previously demonstrated a great activity of Internet users searching for toothache-related issues, with the aim of achieving self-resolution of dental pain through the adoption of alternative methods, such as painkillers and home remedies (*Cohen et al., 2009*; *Ahlwardt et al., 2014*; *Lotto et al., 2017*). These behaviors tend to be exacerbated over the years, since oral diseases and their painful symptoms continue to be highly prevalent worldwide (*Pitts et al., 2011*; *Cohen et al., 2008*; *Peres et al., 2010*), especially among social deprived groups (*Santiago, Valença & Vettore, 2013*; *Da Cunha et al., 2017*).

The preferences of Web health seekers can be influenced by distinct factors, such as the ability of people in obtaining, processing, and understanding information adequately (*Levin-Zamir & Bertschi, 2018*). For instance, participants of Web-based computer-tailored preventive programs preferred videos instead of reading materials, because their usefulness, involvement, and attractiveness (*Walthouwer et al., 2015*). In a similar way, people expect to solve their questions finding adequate health contents in videos online (*Azer et al., 2013*; *Madathil et al., 2015*; *Sahin et al., 2018*). YouTube presents the highest popularity among streaming services, achieving 1.5 billion users and more than 2 billion views per day globally (*Statista, 2018*).

In this sense, hypothetically, source-based patterns of health-related Web searches might differ between individuals from developed and developing countries, due to their social, economic, political, cultural, and psychosocial backgrounds (*Cruvinel et al., 2017*; *Cruvinel et al., 2018*; *Cruvinel et al., 2019*). To our knowledge, there is no evidence comparing the volume of oral health-related searches obtained from search engines and video platforms in different populations. This evidence can contribute to the planning and implementation of educational strategies, guiding the production of contents according to the characteristics of audiences, to minimize the consumption of misinformation. Therefore, this study aimed to determine the trends of toothache-related searches performed by Google and YouTube users, regarding developed and developing countries. The null hypothesis was stated as the levels of toothache-related interests would have no differences between country groups over time ($H_0$).

## MATERIALS & METHODS

### Study design

This retrospective longitudinal study analyzed computational metadata on toothache-related interests of Internet users. Google Trends was accessed to obtain the monthly variation of the relative search volume (RSV) of the topic "Toothache-Disease", considering Google Search (G) and YouTube (YT) from developed and developing countries through

**Table 1 Disability-adjusted life years (DALYs) for countries, country groups and years.**

| Countries | Years | | | | | | | | | |
|---|---|---|---|---|---|---|---|---|---|---|
| | **2008** | **2009** | **2010** | **2011** | **2012** | **2013** | **2014** | **2015** | **2016** | **2017** |
| Australia | 24.67 | 24.62 | 24.60 | 24.68 | 24.91 | 25.27 | 25.74 | 26.31 | 26.95 | 27.64 |
| Chile | 35.09 | 34.80 | 34.65 | 34.60 | 34.58 | 34.56 | 34.54 | 34.54 | 34.55 | 34.57 |
| Japan | 23.59 | 23.33 | 23.19 | 23.15 | 23.10 | 23.06 | 23.04 | 23.02 | 23.04 | 23.09 |
| UK | 36.43 | 36.38 | 36.33 | 36.27 | 36.19 | 36.12 | 36.06 | 36.00 | 35.94 | 35.88 |
| USA | 21.94 | 21.98 | 22.05 | 22.19 | 22.42 | 22.72 | 23.08 | 23.48 | 23.93 | 24.41 |
| **Developed countries** | **28.34**[a] | **28.22**[a] | **28.16**[a] | **28.18**[a] | **28.24**[a] | **28.35**[a] | **28.49**[a] | **28.67**[a] | **28.88**[a] | **29.12**[a] |
| Brazil | 32.05 | 32.01 | 31.99 | 32.01 | 32.02 | 32.03 | 32.05 | 32.09 | 32.14 | 32.19 |
| Mexico | 25.99 | 25.76 | 25.64 | 25.60 | 25.56 | 25.53 | 25.51 | 25.49 | 25.48 | 25.48 |
| Russia | 39.86 | 39.67 | 39.45 | 39.21 | 38.93 | 38.64 | 38.35 | 38.05 | 37.76 | 37.49 |
| Saudi Arabia | 21.95 | 22.10 | 22.22 | 22.32 | 22.41 | 22.48 | 22.56 | 22.65 | 22.73 | 22.80 |
| South Africa | 15.78 | 15.67 | 15.62 | 15.60 | 15.58 | 15.55 | 15.52 | 15.50 | 15.49 | 15.50 |
| **Developing countries** | **27.13**[a] | **27.04**[a] | **26.98**[a] | **26.95**[a] | **26.90**[a] | **26.85**[a] | **26.80**[a] | **26.76**[a] | **26.72**[a] | **26.69**[a] |

**Notes.**
Note that there were no significant statistical differences between DALYs of developed and developing countries over time (similar lower superscript letters, Mann–Whitney $P > 0.05$).

2008 to 2017. The trends, 12-month forecasts and annual ratios of YT/G searches were determined and compared between both country groups.

## Country groups

Ten countries were divided into two groups, according to their socioeconomic development levels. The inclusion criteria were defined as countries from different continents and hemispheres with more than 15 million inhabitants and >50% of Internet penetration (*World Bank Group, 2018*). To avoid selection bias for the comparison of countries, both groups were statistically similar in relation to the burden of untreated dental caries in permanent teeth measured by the disability-adjusted life years (DALYs) (*Institute for Health Metrics and Evaluation, 2018*) (Table 1). Developed countries were composed by Australia, Chile, Japan, United Kingdom, and United States, while developing countries were composed by Brazil, Mexico, Russia, Saudi Arabia and South Africa. This classification is in accordance with *The United Nations (2014)*.

## Relative search volume (RSV)

Google Trends was used for data collection. It depicts results of the monthly variation of RSVs of specific queries, ranging from 0 to 100. These values represent the normalized ratio of search volume of a particular keyword by the overall searches detected in a given time, considering the maximum value of the curve as RSV = 100. Also, results presented by Google Trends can be filtered by location, time, category, and source of information, with the availability of determination of RSVs of popular issues by topics related to predefined and automatic algorithms.

On August 27, 2018, the volume of data related to the topic "Toothache-Disease"—an automatic algorithm that combines all keywords, sentences and terms related to this

issue—was determined according to the activity of Google and YouTube users, through January 2008 to December 2017.

## Data analysis

Data were analyzed with the Statistical Package for Social Sciences (version 22.0; SPSS, Chicago, IL, USA), as follows:

1. RSV trends: the curves of observed RSV values were analyzed heuristically. The trends of toothache-related time series were also checked by autocorrelation (ACF) and partial autocorrelation (PACF) plots.

2. Comparisons between time series: since fitted values of Google- and YouTube-based RSVs were normally and homogeneously distributed (Kolmogorov–Smirnov and Levene tests), T independent test was applied to compare the curves and their segments from developed and developing countries.

3. Forecasting models: data collected until December 2017 were applied to construct 12-month forecasts for toothache-related RSV values. For that, autoregressive integrated moving average (ARIMA) models were chosen by the lowest values of normalized Bayesian information criteria (Normalized BIC), among curves without a significant residual autocorrelation (Ljung–Box test).

4. Comparison of ratios YT/G: since these values were not normally and homogeneously distributed (Kolmogorov–Smirnov and Levene tests), ratios were compared using Kruskal–Wallis and Dunn's test, considering country groups and different years.

For all statistical analysis, $P$ values $< 0.05$ were considered significant.

# RESULTS

## RSV trends

Heuristically, RSVs for Google and YouTube increased in both country groups over time (Fig. 1); however, it was not possible to establish conclusive trends using ACF and PACF analyses (Fig. 2). A significant negative autocorrelation was detected in Google searches from developed countries (lag 2), while a significant positive autocorrelation was detected in Google searches from developing countries (lag 1).

## Comparisons between time series

The percentages of increments of Google- and YouTube-based searches were respectively 146.9% and 156.1% for developed countries, and 288% and 273.3% for developing countries.

The means of Google-based searches on toothache were significantly higher than those of YouTube-based searches in developed (61.74 *vs.* 16.01) and developing countries (46.79 *vs.* 10.91). Comparing the last five years, these significant differences were maintained only in relation to Google-based searches; however, search volumes were similar between both groups in the last two years, whereas YouTube-based searches were significantly higher in developed countries in 2017 (Table 2).
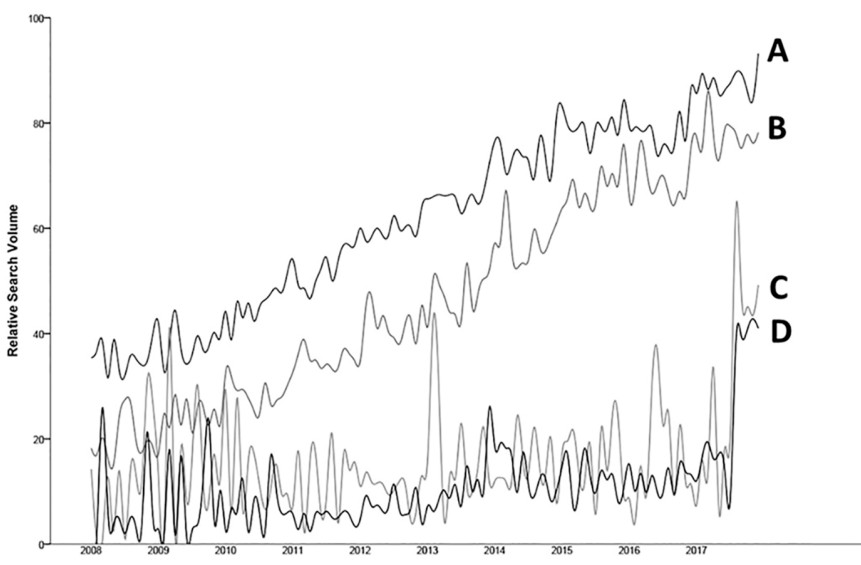

**Figure 1** Relative Search Volume for toothache-related searches in developed countries in Web (A), developing countries in Web (B), developed countries in YouTube (C) and developing countries in YouTube (D).

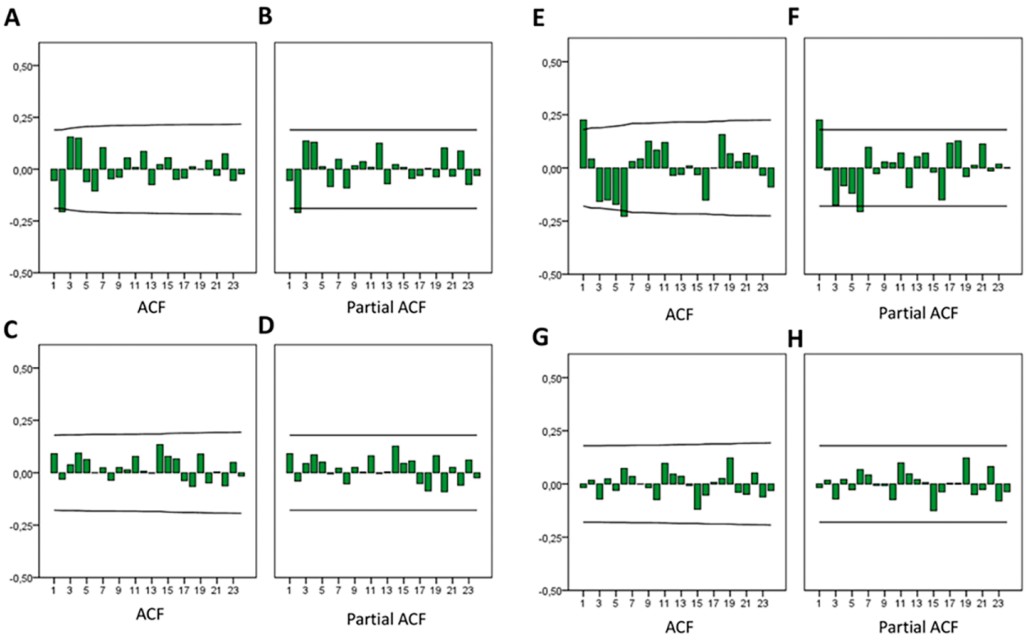

**Figure 2** Autocorrelation (ACF) and Partial autocorrelation (Partial ACF) plots for the monthly variation of RSV toothache. (A/B) developed countries in Web, (C/D) developed countries in YouTube, (E/F) developing countries in Web, (G/H) developing countries in YouTube.

**Table 2 Mean (±SD) of toothache-related searches performed in Google and YouTube for users from developed and developing countries in different periods.**

| Period | Google searches | | YouTube searches | |
|---|---|---|---|---|
| | Developed countries | Developing countries | Developed countries | Developing countries |
| **Whole period (2008–2017)** | 61.74 ± 5.80[Aa] | 46.79 ± 20.35[Ba] | 16.01 ± 5.56[Aa] | 10.91 ± 5.88[Ba] |
| **Last 5 ys. (2013–2017)** | 77.16 ± 3.52[Ab] | 63.77 ± 11.75[Bab] | 19.37 ± 6.19[Aa] | 14.95 ± 5.97[Aa] |
| **Last 4 ys. (2014–2017)** | 79.91 ± 2.84[Ab] | 67.92 ± 8.32[Bb] | 19.81 ± 7.06[Aa] | 15.94 ± 6.40[Aa] |
| **Last 3 ys. (2015–2017)** | 81.86 ± 2.91[Ab] | 71.43 ± 5.48[Bab] | 21.42 ± 7.69[Aa] | 16.71 ± 7.61[Aa] |
| **Last 2 ys. (2016–2017)** | 83.01 ± 4.64[Aab] | 73.29 ± 6.27[Aab] | 23.70 ± 9.33[Aa] | 18.89 ± 9.36[Aa] |
| **Last year (2017)** | 87.84 ± 2.50[Ab] | 77.78 ± 3.17[Ab] | 30.30 ± 19.16[Aa] | 25.50 ± 13.47[Ba] |

**Notes.**
Upper superscript letters indicate significant statistical differences between country groups considering the same engine search. Lower superscript letters indicate significant statistical differences between periods considering the same country group and engine search.

**Table 3 ARIMA model fit statistics.**

| Country group Web platform ARIMA model | Normalized BIC | MAPE | Ljung-Box | Model parts | Lag | Estimate | SE | p |
|---|---|---|---|---|---|---|---|---|
| Developed Countries Google | | | | | | | | |
| (0,1,1)(0,1,1) | 2.42 | 3.91 | 0.77 | MA | Lag 1 | 0.60 | 0.07 | <0.01 |
| | | | | MA, Seasonal | Lag 1 | 0.53 | 0.08 | <0.01 |
| Developed Countries YouTube | | | | | | | | |
| (0,0,0)(0,0,0) | 4.19 | 113.96 | 0.83 | Constant | | 13.36 | 0.70 | <0.01 |
| Developing Countries Google | | | | | | | | |
| (0,0,0)(1,1,0) | 2.89 | 6.50 | 0.40 | Constant | | 0.16 | 0.01 | <0.01 |
| | | | | AR, Seasonal | Lag 1 | −0.48 | 0.08 | <0.01 |
| Developing Countries YouTube | | | | | | | | |
| (0,1,1)(0,0,0) | 2.49 | 32.54 | 0.52 | Constant | | 0.10 | 0.01 | <0.01 |
| | | | | MA | Lag 1 | 1.00 | 0.04 | <0.01 |

## Forecasting models

Table 3 summarizes adequacy measures and parameter estimation of forecasting models for toothache-related RSV values. The excellent adequacy of ARIMA models was demonstrated by low values of normalized BIC (2.42–4.19). The curves of observed and fitted values of ARIMA models are showed in Fig. 3. Twelve-month forecasts tended to plateau, independently of the source of information and country groups.

## Ratio between Google and YouTube searches (YT/G)

The volume of Google-based searches was higher than that observed in YouTube between 2008 and 2017. Additionally, there was no statistical difference between the ratios of YT/G searches over time, considering both country groups (Fig. 4).

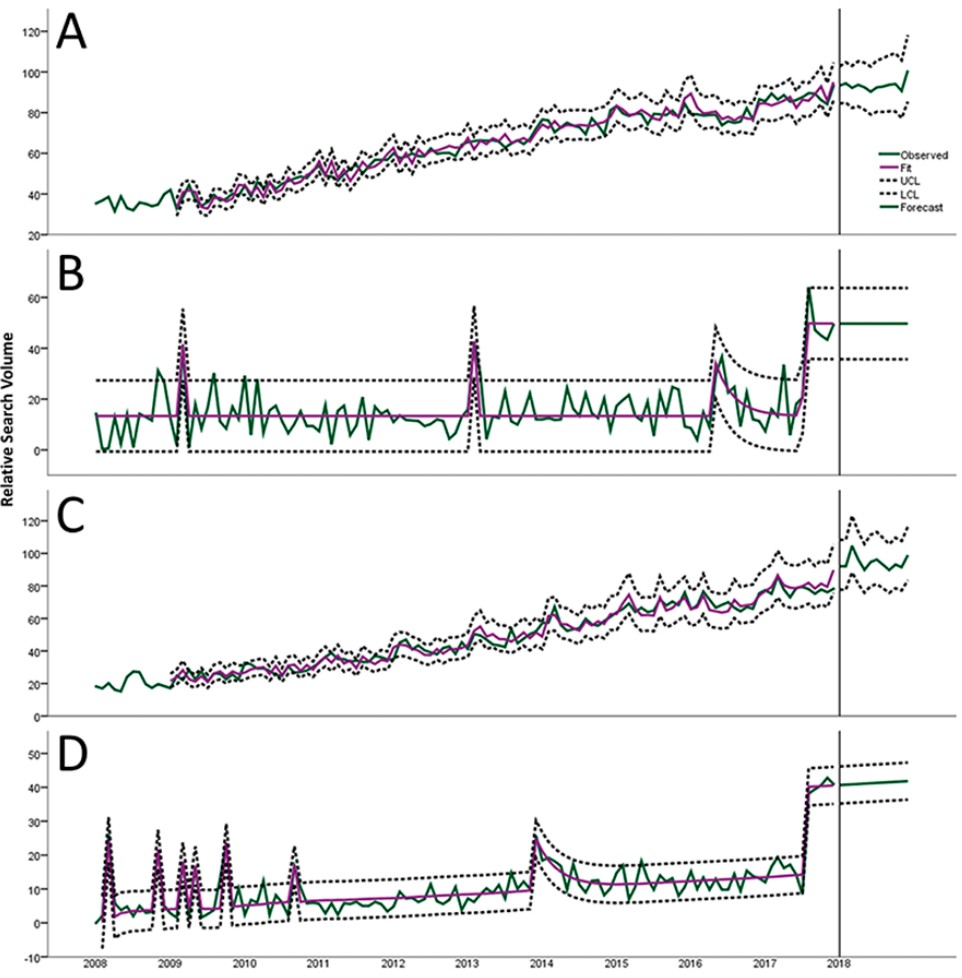

**Figure 3** Predictive charts demonstrate the variation of RSV in developed countries in Web (A), developed countries in YouTube (B), developing countries in Web (C) and developing countries in YouTube (D). Note that RSVs presented after 2018 represent 12-month predictive values.

## DISCUSSION

These findings indicate the increment of volumes of toothache-related searches through 2008-2017, with differences between developed and developing countries. The volumes of Google- and YouTube-based searches were greater in developed countries, with differences decreasing for Google and increasing for YouTube in recent years. Also, the ratios YT/G remained constant over the years, independently of country groups. To our knowledge, this is the first study that compared the interests of Google and YouTube users in toothache-related information, regarding the socioeconomic development levels of countries.

The significant growth of toothache-related interests observed in developing countries in the last years can be explained by the increase in their Internet penetration. This phenomenon can be explained by three main factors: the rapid expansion of digital infrastructure, the popularization of gadgets, and the decrease of Web access costs (*World Bank Group, 2018*;

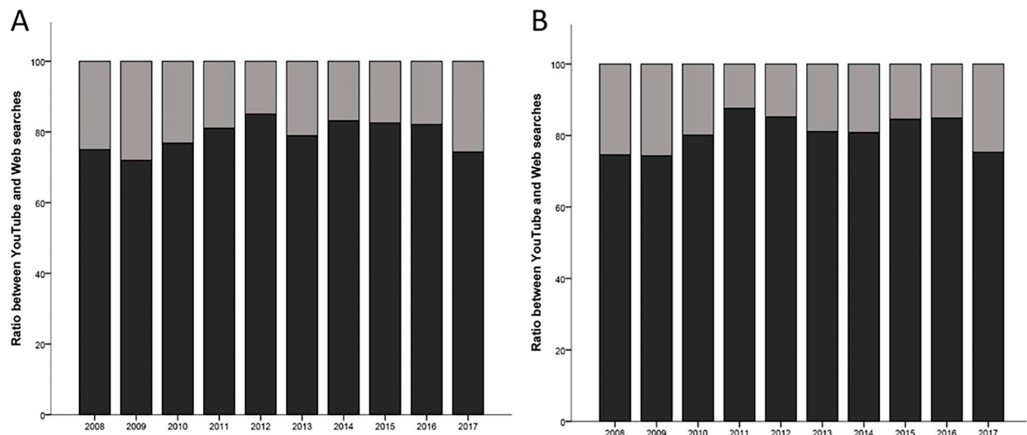

**Figure 4** Ratio between YouTube and Web searches for developed countries (A) and developing countries (B).

*Alliance for Affordable Internet, 2018*). An interesting set of results was detected, as follows: (i) the volumes of Google-based RSVs were maintained significantly higher in developed countries only until 2015; (ii) distinctly, the volumes of YouTube-based searches were similar between both groups from 2013 to 2016; (iii) in 2017, YouTube-based searches returned to be significantly higher in developed countries, as noted in the whole period of time (2008–2017). These findings may be related to the late digital inclusion of developing countries, since netizens are more prone to access health information for solving their own conditions in comparison to new Internet users (*Perrin & Bertoni, 2017*). Secondly, lower levels of education can influence the preferences of users in relation to Web media, favoring the consumption of videos instead of reading materials (*Walthouwer et al., 2015*). This pattern could explain the similarities of YouTube-based searches between both country groups. Finally, a new YouTube-based difference found in the last year of analysis can be related to the mobile access to the Internet, which differs in diffusion, cost and quality according to countries (*Fox & Duggan, 2013*).

The growing interest in toothache observed in both platforms can be justified by the negative impact of untreated dental caries in permanent teeth on the quality of life of individuals (*Institute for Health Metrics and Evaluation, 2018*). This behavior is related to the self-resolution of oral symptoms (*Cruvinel et al., 2019*), motivated by barriers for dental treatment, such as high expenses of private services and insufficient offering of public attendance (*Dhama et al., 2016*). The choice to use YouTube as a source of health information is explained by the rapid increase of its market share, with an annual increase of approximately 17% in audience projected through 2021 (*Statista, 2018*), and by the enhancement of learning that it provides for Web users (*Clifton & Mann, 2011*; *Barry et al., 2016*; *Shatto & Erwin, 2017*).

Some methodological aspects deserve more clarification. Data were collected from Google Trends because it enables the comparison of search volumes of specific topics between Google and YouTube, both leaders in their markets (*Net Market Share, 2018*).

Also, the selection criteria of countries considered a minimum population and Internet penetration to provide an adequate amount of data for statistical analysis. As a result, the country groups presented similar DALYs indices, avoiding the influence of the burden of dental caries on the interpretation of these results. Moreover, the geographic diversity of countries tended to minimize the influence of social, cultural and economic factors during these analyses.

These results need to be interpreted with caution. These data are referring exclusively to the activity and behavior of users from two Web platforms, without considering information from other search engine tools. Additionally, it is not possible to associate socioeconomic characteristics of particular users with their behaviors, since their searches are performed anonymously. Also, this approach can overestimate the activity of netizens, because of the impossibility of excluding duplicate searches performed by the same person in two or more devices. Finally, although behaviors of users lead us to believe that growing searches are related to self-resolution of oral diseases, it is not possible to state that this always translates into the use of remedies or the adoption of alternative measures.

## CONCLUSIONS

In conclusion, toothache-related searches performed on Google and YouTube increased independently of the development levels of countries. The media preferences of users seemed to be influenced by the differences in the education levels, spreading and availability of the Internet found in developed and developing countries. Therefore, the hypothesis $H_0$ was rejected. Indeed, the analysis of these computational-based data suggests that subsidizing health educational programs and campaigns can empower people to make correct and autonomous oral health-related choices. This strategy is important to minimize damages caused by the ineffective self-management of toothache, which can lead to dental loss and systemic complications.

### Funding

This study was supported by the São Paulo Research Foundation with a master's research scholarship (#2017/25899-7). The funders had no role in study design, data collection and analysis, decision to publish, or preparation of the manuscript.

### Grant Disclosures

The following grant information was disclosed by the authors:
São Paulo Research Foundation with a master's research scholarship: #2017/25899-7.

### Competing Interests

The authors declare there are no competing interests.

## Author Contributions

- Matheus Lotto and Thiago Cruvinel conceived and designed the experiments, performed the experiments, analyzed the data, prepared figures and/or tables, authored or reviewed drafts of the paper, approved the final draft.
- Patricia E.A. Aguirre performed the experiments, authored or reviewed drafts of the paper, approved the final draft.
- Anna P. Strieder and Agnes F.P. Cruvinel authored or reviewed drafts of the paper, approved the final draft.

## Data Availability

The raw data is available as Supplemental File.

## Supplemental Information

Supplemental information for this article can be found online at http://dx.doi.org/10.7717/peerj.7706#supplemental-information.

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
