# Peer review of "Levels of toothache-related interests of Google and YouTube users from developed and developing countries over time"

_PeerJ, doi:10.7717/peerj.7706_

## Round 0.1 · original submission · Major Revisions

While the results of your manuscript support the conclusions, your manuscript can be improvised in several areas. English language usage would be one area. The usage of language appears to be casual and the paragraphs are broken into smaller sections which breaks the readers' thought process. Another area of improvement would be to address the reviewer comments. Why the specific key word "Toothache-Disease" is used for the search criteria as opposed to words like "Toothache" or "Tooth pain" or "Dental pain" needs to be addressed. Another aspect that has not been addressed properly with valid references is your definition of developed and developing countries.

Reviewer 1 ·

Basic reporting

The article falls below standard English usage in both style and grammar. At various places, it is difficult to make sense of the manuscript. Also, the manuscript does not flow well and it is not an easy read.

The manuscript is sufficiently supported by literature, however the details presented in the manuscript does not support the overall context well.

Figures and other data are presented well. Raw data excel file is missing some information.

Results are relevant to hypothesis but the hypothesis itself is not very informative.

Experimental design

Methods are fairly well laid out but the utility of multiple testing to largely look at trends of usage of a keyword does not provide an interesting view of the problem and therefore conclusions are restrictive.

Validity of the findings

There seems to be little impact of the results in the overall field of data science relevant to user interests of data platforms for questions relevant to dentistry.

The discussion of the manuscript is informative and brings out relevant points but the remaining article does not dissect the topic in detail. While I understand that this is restricted by data available from Google trends, the information reviewed and presented contributes little to the advancement of the topic.

Reviewer 2 ·

Basic reporting

Interesting study and relevant to current scenario

Experimental design

Abstract:
Lines # 40-42: Background: Authors have mentioned “In this sense, hypothetically, source-based patterns of health related Web searches might differ between individuals from developed and developing countries, due to their educational backgrounds.” Differences could be due to broader social, economic, political, cultural, psycho-social reasons, not just educational backgrounds. The authors can consider rephrasing this sentence.

Line # 46: Methods: “This longitudinal study analyzed……” Methodology section of main text indicates that this is a “retrospective longitudinal study”. The latter is more appropriate.

Main text:
Line # 64: Introduction: “The spreading of Communication and Information Technologies (ICTs) stimulated individuals in…..”. Consider changing “Communication and Information Technologies” to “Information and Communication Technologies”, as indicated in the parenthesis by the authors
Line # 104: Search term included was “Toothache-Disease”. Individuals might use many other search terms/key words while using the internet to know about tooth ache.
Lines # 110-112: “Developed countries were composed by Australia, Chile, Japan, United Kingdom, and United States, while developing countries were composed by Brazil, Mexico, Russia, Saudi Arabia and South Africa”. Was this classification based on World Bank, United Nations or any such agencies? Kindly include reference for the same.
Lines # 110-112: Among the countries included, notable exceptions are China and India. Can authors provide any explanation for the same?
Lines # 110-112: Can the authors provide actual information about “availability and penetration of the Internet” in these countries? It will enhance the value of the findings of the present study.
Lines # 112-115: “To avoid bias related to the selection and comparability of countries, both groups were statistically similar in relation to burden of untreated dental caries in permanent teeth, characterized by toothache (Institute for Health Metrics and Evaluation, 2018).” Can authors present the results in tabular form and explain what “statistically similar” is?
Lines # 110-112: The usual reference cited for global burden of disease is World Health Organization (WHO) databank. Can the authors explain why or how Institute for Health Metrics and Evaluation, 2018 provides greater/better data than WHO?
Lines # 110-112: Untreated dental caries need not always present itself as toothache. Can the authors make a note of this in the discussion section?

Validity of the findings

Results:
Table 2 can be presented as first table. Can the authors provide descriptive explanation of their findings before presenting RSV trends? They can inform the reader about mean Google and YouTube searches for developed and developing nations.

Can the authors consider exploring intragroup comparisons of outcomes? It would be interesting to observe for a statistically significant increase in the number of searches in Google and YouTube within the 2 groups.

Discussion:
Internet usage may not necessarily translate into initiation of remedial or corrective measures. Authors can consider adding the same in discussion section

---

## Round 0.2 · accepted · Accept

You have addressed the issues raised by the reviewers in a considerable manner. Your study is unique in that it provides the methods for extracting information from platforms like Google and YouTube.

Reviewer 2 ·

Basic reporting

Nocomment

Experimental design

Corrections done

Validity of the findings

Relevant

Additional comments

Can be accepted